# A Study on Parenting Experiences of Multicultural Families with Disabled Children in Korea

Keoungyeol Kim

Department of Social Work, University of YoungSan, Busan City 48015, Korea; 007powerk3@hanmail.net

**Abstract:** This paper employs a qualitative case study to investigate the parenting experiences of multicultural families with disabled children in Korea. The topic at hand focuses specifically on mothers' experiences of raising a child with disability in a multicultural family. Participants chosen through purposive sampling method were recommended by the Korean multicultural support center. This study's five participants are married female immigrants who are raising elementary school-age children with disabilities and who have Korean communication skills. The qualitative case study approach seeks to elicit the participants' experiences of child nurturing. Their experiences were analyzed and categorized into two main categories: (1) hardships for mothers raising children with disabilities and (2) expectations of mothers raising children with disabilities. Such analyses lead us to understand the intricacies of motherhood for children with special needs in foreign countries. Lastly, the implications of this study aim to provide direction for effective and practical policies—including social welfare and educational support—that will meet the needs of multicultural families with disabled children.

**Keywords:** multicultural society; social welfare; multicultural education; social welfare for multicultural individuals

## 1. Preface

The number of multicultural families in the U.S.A. is expected to increase by 213% between 2000 and 2050, by which time, it is thought that around 8% of the population of the country will be Asian Americans (Wang and Casillas 2013). Similarly, South Korea is also rapidly transforming from an ethnically homogeneous society to a multicultural one (Kim and Hwang 2019). Korea is developing into multicultural society as foreigners migrate to Korea for a variety of reasons such as marriage, employment, and education. The country's transformation into a multicultural society first began in the early 1990s with the influx of foreign workers into domestic industrial positions. Additionally, in the mid-1990s, the Korean government initiated a campaign to marry young men in rural areas with women from underdeveloped countries (Kim et al. 2012). According to 2018 Statistics Korea Population and Housing Census, the number of multicultural households in Korea has reached 335,000 accounting for 2% of the total population. To prepare for such an increase in multicultural families, the 2009 Multicultural Families Support Act was enacted to provide an institutional framework for improving the quality of life of multicultural families through social integration and household stability (Han 2018). Since 2012, when the multicultural student survey began, it has been found that the ratio of multicultural students is increasing every year at all school levels. In particular, elementary schools with the highest proportion of multicultural students increased 3.6 times from 1.1% in 2012 to 4.0% in 2020; middle schools and high schools increased by 4.0 times and 4.5 times compared to their 2012 values, respectively (Lee 2019). To support the school lives of students from multicultural families, the government has prepared the 'Student Education Support Plan for Multicultural Families'. A government survey on multicultural families that indicated that 6.4% of multicultural households have a member with a disability, and

about 10% of all households with a special needs member are also multicultural families (Lee 2019).

Families with children with disabilities were found to experience hardship twofold in both education and medical care as well as more serious social prejudice and discrimination. The government continues to expand policies and support for multicultural families, but multicultural families raising children with disabilities or children at risk are not provided with services suitable to meet their needs within the current framework of multicultural family support. In addition, immigrant families point out that the health care services in the United States, as well as services for children with disabilities, are often better than those in their home countries (Lee et al. 2022) It is therefore crucial first to understand and second to improve the current framework of multicultural family support so that their needs may be satisfied. A qualitative approach is appropriate for such research because this allows for the examination of the "essence" of lived experiences. Current qualitative studies are limited in that they focus on the experiences of raising able-bodied children of multicultural families (Boratav et al. 2021; Chen et al. 2021; Song et al. 2016). Other research has been done on the difficulties faced by marriage immigrants or their successful adaptation (Ataca and Berry 2002; Sigad and Eisikovits 2009). Fortunately, some quantitative studies have considered the experiences of multicultural families raising children with disabilities (Lopez et al. 2018; Wang and Casillas 2013). Although previous attention has been paid to mothers raising children with disabilities (Lopez et al. 2018; Song et al. 2016), research on raising children with disabilities from multicultural families living in Korea is rarer still. This present study finds its motivation in the aforesaid literature gap with a vision to provide a fundamental understanding about multicultural motherhood of children with disabilities in South Korea.

Generally, mothers in Korea have the primary responsibility to raise their children, and thus married immigrant women in Korea are responsible for raising children as well. However, married immigrant women experience specific challenges raising their children, challenges which arise from the linguistic and socio-cultural environment. Apparently, such challenges are greater for married immigrants from multicultural families raising children with disability. Research has demonstrated that immigrant parents face additional barriers affecting their access to, utilization of, and experience with health care; consequently, this leads them to be less engaged in their child's treatment, resulting in poorer rehabilitation outcomes (Norheim and Moser 2020; Turney and Kao 2009). An immigrant parent may face a variety of stressful situations such communication in an unfamiliar language, financial insecurity, and acclamation to a new culture. Studies demonstrated that there is a lack of understanding about a child's disability among immigrant parents due to language difficulties. In this study, the disabled are limited to those with developmental disabilities. In Korea, the category of developmental disability is regarded as intellectual disability and autism according to the Developmental Disability Support Act. In this study, the area of people with developmental disabilities is limited to intellectual disabilities and autism disorders. Intellectual disabilities and autism disorders are considered disabilities, not diseases, in Korea. Intellectual disability and autism disorder are types of cognitive disability, and unlike people with physical disabilities, they have cognitive limitations and difficulty in adaptive behavior. The common point between cognitive disabled and physically disabled is that they require environmental changes such as reasonable acccommodation, and that special education services can be received during school age as special education subjects.

Upon encountering such life contexts this author has become interested in the field's understanding of raising children with disabilities from the perspective of multicultural marriage immigrants. By examining their acts of child rearing, the need for additional research on multicultural marriage migrants' experiences was raised. Therefore, the purpose of this study is to examine the experience of mothers raising children with disabilities in multicultural families. With research studies such as this, it is possible to explore the reality of maternal child rearing in multicultural families as well as new directions and applications of policy in social welfare and educational support. The guiding research question for this qualitative case study is, "What is the experience of a mother in a multicultural family

who is raising a child with disabilities in Korea?" Even if this study is limited to the cases in Korea, it is expected that insight can be used for cases in other similar cultures where mothers have the primary roles in raising children.

## 2. Research Methodology

The purpose of this study is to explore the experiences of marriage immigrants who are the main caretakers of children with disabilities in multicultural families. More precisely this study investigated a number of challenges that marriage immigrant encounter in educating their children with disabilities, as well as their needs from general society and the Korean government to overcome these challenges.

Generally, case studies on the lives of mothers raising children with disabilities in multicultural families are very limited, and the Korean research field is similar. Fortunately, the case study method provides useful tools in a qualitative research when speciality instances of a concept exist. Since the experience of marriage immigrants as the main caretakers of children with disabilities in multicultural families is not a widely established research field, a qualitative case study method has the significant potential to stimulate new insights.

### 2.1. Participants

The interview data generated from the study's five questions created a limited sociodemographic overview of the participants. Most mothers come from Asian countries geographically close to Korea, and many came to Korea in their 20s. Generally, they had their children within a few years of their arrival. All mothers have lived in Korea for more than ten years, and currently they are living in a medium-sized city. Their elementary school-aged children's disabilities are primarily non-physical. Their specific educational backgrounds are unknown, but the data seems to indicate that they have completed the equivalent of a high school diploma.

Although female marriage immigrants face hardships in obtaining jobs, engaging with society, and communicating in Korean, they do have some measure of support. At the national level, the 2013 Multicultural Families Support Act, 2014 Protection and Promotion of Cultural Diversity Act provide enforceable legal protections against unwarranted hardships. Additionally, the mothers in this study have access to social services through local welfare centers as well as medical services through national health insurance plans. It remains to be seen, though, if their access to such services is quick and easy.

A purposive sampling method was used to select the research subjects. Purposive sampling was appropriate here for recruiting participants who can provide in-depth and detailed information about the phenomenon under investigation. This was done upon the recommendation of officials from the Multicultural Support Center and Office of Education Support Center in a medium-sized Korean city. The study participants were married immigrants who are raising children with disabilities in multicultural families. The participants were limited to five mothers of multicultural families who had elementary school-aged children with disabilities and who had the ability to communicate in Korean. The purpose and process of the study were explained to all five participants. Table 1 summarizes key data about the study participants.

**Table 1.** Research Participant Data.

| Participant | Gender | Age | Nationality | Time in Korea | Age of Child | Child's Disability |
|---|---|---|---|---|---|---|
| Participant 1 | female | 38 | China | 15 years | 12 | mental retardation |
| Participant 2 | female | 35 | Australia | 14 years | 11 | mental retardation |
| Participant 3 | female | 40 | Vietnam | 11 years | 9 | mental retardation |
| Participant 4 | female | 29 | Vietnam | 13 years | 10 | mental retardation |
| Participant 5 | female | 35 | Philippines | 15 years | 12 | autism |

*2.2. Interview Questions*

This study aims to explore the experiences of marriage immigrants. Structured interviews included the following five questions:

1. How is your life in Korea as a female marriage immigrant?
2. When did you first learn that your child had a disability?
3. What was the hardest time you had raising your child?
4. What kind of educational support do you think is needed for your child's education?
5. Do you have any plans for your child's education in the future?

*2.3. Data Analyses*

Data were collected through an interview method to reveal the subjective nature of the participants from an insider's point of view. Interviews were conducted from March 2021 to April 2021. All contents were recorded with the consent of the interviewee, and the recorded material was transcribed. In order to conceptualize the transcribed original data, read carefully to grasp the meaning of words and sentences, and conceptualize them as an integrated work. In each case, related concepts were grouped into one category to explain the characteristics and scope. At the beginning of the interview, participants were asked general questions about their lives. The purpose of the study was explained to the study participants, informed consent was obtained, and the interview was conducted. Interviews were conducted in a work lounge or café setting, both familiar spaces where the study participants may talk comfortably.

**3. Results**

The central concept arose through the discovery of contextual meaning in how the study participants raised multicultural children with disabilities and organizing ideas into themes and subthemes. The researcher continued to compare cases and find commonalities and differences. After consideration and interpretation of the data, 18 subcategories combined to form two final categories. Table 2 visualizes the data on the parenting experiences of female marriage immigrants.

In the following Sections 3.1 and 3.2, we summarize the hardships for mothers raising children with disabilities and educational hardships. Each of them are followed by more specific subjects. This discussion includes actual interviews with participants. We begin our discussion with hardships for mothers raising children with disabilities.

**Table 2.** Inter-case analysis of female marriage immigrants.

| Subcategories, Main Categories | | Final Categories |
|---|---|---|
| Personal guilt about giving birth to a child with a disability | Social and emotional hardships | hardships for mothers raising children with disabilities |
| Social perspectives on people with multicultural backgrounds | | |
| Social perspectives on people with disabilities | | |
| Lack of Korean language education | Educational hardships | |
| Lack of a support system | | |
| Distrust of the Korean education system | | |
| Child care services | Mothers' expecations for Korean society | Expectations of mothers raising children with disabilities |
| Support group availability | | |
| Support for out-of-school multicultural children | | |
| Bilingual services | | |
| Universal environment | | |
| Strengthening Korean language education | Mothers' expectations for their children's education | |
| Special education teachers and their multicultural competencies | | |
| Differentiated learning | | |
| Establishment of a support system | | |
| Developing an alternative evaluation system | | |
| Vocational education reinforcement | | |
| Curricular and extra-curricular activites | | |

*3.1. Hardships for Mothers Raising Children with Disabilities*

3.1.1. Social and Emotional Hardships Bulleted Lists Look like This

A.     Personal guilt about giving birth to a child with a disability

This study's participants are housewives of multicultural families who are under heavy stress while raising children with disabilities. One mother in particular said she was living a difficult life every day, feeling guilty about giving birth to a child with a disability who must bear the weight of such a life. Another participant shared that she felt very sorry for her child when she heard the doctor's opinion that the disability was caused by the stress that her child experienced during pregnancy.

The following are detailed statements of two interviews.

My child was assessed as being in grade 3, and we also found out she has ADHD. We don't want other children to suffer because of our children. We decided to try a drug treatment. There was a noticeable effect: my child takes the medicine and then becomes extremely quiet. When my child is sick, she loses her energy, and all I see her doing is lying down and sitting still. I thought it was okay to stop taking the medicine, so I consulted with the doctor in charge. It was difficult for her to keep taking psychiatric medication for more than 2 years. It hurts my heart to think that my child has a disability because of me. As a parent I wonder what will happen to my child with a disability when I die? It hurts me deeply to think about this. (Participant 1)

When my child was two years old, he couldn't walk or speak well, so I took him to get a disability test and a height test. My baby started walking at the age of three, but he wasn't talking at that time; in fact, he's still a bit mute now, so he's in speech therapy. The doctor told me that the stress I had when I was pregnant had an effect on the baby in my womb, so it seems that I'm the one with a disorder. Upon hearing the doctor's words, I felt sorry for my child. (Participant 3)

B.　Social perspectives on people with multicultural backgrounds

Having lived in Korea for a long time, the prejudice and discrimination experienced by female marriage immigrants made their lives more difficult. In particular, one participant said that her self-esteem was greatly hurt because of Koreans who recommended her to buy cheap items. Also, some study participants complained about their children being discriminated against as multicultural individuals. One mother stated that since her children were born and raised in Korea, she would like them to be recognized as Koreans rather than multicultural people.

The following are detailed statements of two interviews.

When my child is doing something in school, there are certain benefits. In general, multicultural families have priority. I really wish I hadn't done these things. These children are Koreans who were born in Korea and raised in Korea. But I have to say that it is a bit discriminatory because we keep adding the word multiculturalism. It's just the family that is multicultural: the child is just a Korean child, and there are various cultures in this family. There's the mothers culture and the father's culture. I don't like hearing things like that. What about the multicultural children themselves? (Participant 1)

Sometimes when I went to the supermarket, I was discriminated against because I was a foreigner and didn't seem to have much money. One time I was going to buy a cell phone, a Samsung one, but I went to an LG shop. I asked the clerk who then replied "You can't buy this phone, it's not available at our store. You have to go to Samsung, but I can't buy this phone because it's expensive." Another time I was trying to buy clothes again, but I was told, "This dress is expensive, you can't buy it (pointing to other clothes). This dress is five thousand won. It's ten thousand won. Buy this." (Participant 4)

C.　Social perspectives on people with disabilities

The study participants' children experience double social discrimination: first as multicultural people and second as people with disabilities. In particular, when seeing children who are often bullied at school and living alone, the study participants said that their hearts became heavy. One study participant, seeing her child being rejected at an afterschool private learning academy, expressed her hope that Korean society would broaden its understanding of people with disabilities. She hopes that the environment in which abled people view people with disabilities as socially segregated will change as soon as possible. She is also preparing to immigrate to another country where there is no discrimination against people with disabilities.

The following are detailed statements of four interviews.

When I my child entered elementary school, in the first and second grades, all the kids all played together; but, once they were promoted to higher grades, it was difficult for my child to get along with other children of his age. It's hard to socially mingle and talk with classmates, so even they can't have any conversation. Because they recognize him like this, they don't hang out with him. So when my child went to high school, he was alone and went to school alone. He couldn't play with anyone. So he must have been hurt in his own way. I always ask him, if he's getting along well with his classmates, but he just say things like "No we didn't do anything together." (Participant 1)

To be honest, there are a lot of things that the neighborhood welfare centers do. I do think that services for people with disabilities are quite lacking. It would be nice if schools or non-disabled people could broaden their minds about people with disabilities, but I don't think there's a way to solve that. Here (in Korea), people with disabilities are classified as completely disabled; they work separately, so they are on their own. I think they don't find their place in society because they live separately from it. (Participant 2)

My child is currently attending a special education school. She is alone, not hanging out with other children. She likes to play games on the phone or computer, especially in the elevator. She likes elevators so much that she doesn't move when she sees them. (Participant 3)

Even now, I am not good at Korean. When I first came to Korea, my mother-in-law was suffering from a chronic disease called dementia, and my child also had a disability, so I could not study Korean at all while I was staying at home for three years. I was given the opportunity to study Korean at the welfare center, but within a year of starting it, I had a second child, so I quit. After that, when my children had free time to go to school, I studied Korean again. Now, I am working. I can't speak Korean well, so I get a strange feeling from people around me. (Participant 4)

### 3.1.2. Educational Hardships

A. Lack of Korean language education

Participant recognizes that their children's Korean language skills are the basis for learning, and they want their children to improve their Korean language skills through after-school Korean language classes compared to non-multicultural people.
The following are detailed statements of one interviewee.

At school, there is only one program that I study; I do not do anything else after school. There is also is no Korean study room. There is only one course to help people study Korean (in a Korean Language department). What I want is to have a separate Korean language study room. The learners are not only foreign children but also Korean children who do not know the language well at first. (Participant 4)

B. Lack of a support system

The participants in this study argued that, for the education of multicultural children with disabilities, the educational environment should be actively supported and the level of educational goals should be lowered. One mother said that Korean education provides the same standards without considering the level and ability of students, and that reasonable measures are required for success. Her opinion is that education should be supported by focusing on language education in the lower grades and focusing on vocational competency education in the upper grades.
The following are detailed statements of two interviewees.

First, you need to give children support. Especially language support, as much as possible, with high expectations. In fact, in the long run, your child should get good grades in school. Even if you go to a good college or not, eventually you have to become an adult and get a job in society to make a living. To help with that, looking at the long-term is important. (Participant 1)

Give lots of language support when they are young, Lower their expectations so that they can succeed at their own level. After that there should be a, for example, program for new high school students that allows you to gain other skills when preparing for college. All with lower expectations than the general public. We have to see this plan to the end. If this child fails in school and if there are no future opportunities, then he enters society as a burden to us. (Participant 2)

C.　　Distrust of the Korean education system

Study participants with abundant experience living abroad expressed distrust in the knowledge-based memorization method. Rather than sending their children to school, they expressed their wish to homeschool.

The following are detailed statements of two interviewees.

I wanted the education to be conducted with a focus on the students' interests rather than on an environment where students only sit on chairs. From our point of view as parents, we lack social skills (the ability to making friends), so at that time it would be better for the child to be in the class. And frankly, helping my child out one-on-one with homework for the purpose of going out and studying (for a special class), it doesn't feel very helpful. I think homeschooling would be better from an educational point of view. Or children can learn through private tutoring (hagwon), but honestly, I don't think schools do anything. (Participant 2)

Education varies depending on the teacher, but I think it's okay to do it with another teacher because my child's teacher has changed this time. The biggest reason is that the teachers tell us that students whose grades drop beginning in the 4th grade need to have additional classes after school. It's a shame for my child if you attend special education classes, you don't get these extra lessons. Additional classes are required to raise grades and cannot be selected, so we have no choice but to put our children back into the special class. In the case of special classes (regular classes alone), children do not take additional classes because they are already tired. (Participant 5)

### 3.2. Expectations of Mothers Raising Children with Disabilities

3.2.1. Mothers' Expectations for Korean Society

A.　　Child care services

The female marriage immigrants who participated in this study wanted the government services that support their children and families to expand. One mother in particular wanted the state to make efforts to ensure that children universally receive child care services during infancy and childhood, regardless of their families' wealth. This is because there is an expectation that children can learn Korean well through the Korean people who are providing care services rather than through the mothers who are not fluent in Korean.

The following are detailed statements of three interviewees.

What I felt while raising children with disabilities is that multicultural migrant women give other mothers a lot of support once they get married and enter the country. In terms of education, there are many things that are done with the purpose of quickly adapting to [Korean] society and living happily. For children, there are developmental therapy services because there are many children with slow development in multicultural families. Other than that, I don't think there is much support. Then mothers need education too. But for the kids, it's nonsense that the mother is her primary caregiver and that the mother raises her child with that clumsy language while she teaches the child Korean. So, as I said before, I would like to see more expansion of childbirth assistance services and childcare services. (Participant 1)

I wish I had someone who could teach my children at home. In the past, volunteers came and taught me for a while, but I thought it would be nice to have a teacher like that at home. I wish there were visiting teachers for the children. (Participant 3)

Caring service teachers come to play with the children, and all of those teachers visit the house mainly for the children. I hope that a teacher like that will come to play with the child, feed them, give them baby food, play with them a lot,

and create an environment like that so that they can listen to a lot of Korean. (Participant 5)

B.    Support group assistance

It is difficult to live as a female marriage immigrant in Korean society, but the life of a female marriage immigrant with a child with a disability is even more difficult. Some participants indicated their desire for national or local governments to create gatherings for mothers in the similar life situations. There they could meet each other to exchange information and to support and encourage each other.
The following are detailed statements of two interviewees.

I wish there were teachers who visited children with disabilities, and I also hope that mothers could receive counseling. Mothers of children with disabilities are under a lot of stress. In particular, multicultural mothers are very stressed because they do not understand and do not know much [about Korea or Korean culture], so I want mothers of multicultural families to pay more attention as well. (Participant 3)

I hope that mothers with children with disabilities are not ashamed to talk and share with each other and to have children with disabilities. I think these negative thoughts should go away. (Participants 5)

C.    Support for out-of-school multicultural children

The study participants showed a lot of interest in the educational rights of multicultural children who are in a similar situation to their own children. In particular, among multicultural children, they suggested that social welfare centers and multicultural support centers in the local community should pay attention to out-of-school children and adolescents even if they did not adapt well to the formal, institutional academic environment.
The following are detailed statements of one interviewee.

I don't know very much about elementary school life. First of all, in the case of our children, if you look at multicultural children, of course, there are children who go to middle and high school with their peers. However, there are still a few children who do not go to middle school, and there are many children who wander about society. I hope that social workers will pay a little more attention to these children and connect them with a lot of vocational mass-production education, so that they can learn a skill for future work. (Participant 1)

D.    Bilingual service

Some study participants wanted the state to support their children to study not only Korean but also their mother's native language. Through this, it is expected that mothers and children will be able to communicate smoothly and grow into adults with a multilingual international awareness.
The following are detailed statements of one interviewee.

I want my children to learn not only Korean but also Vietnamese. I want the centers and schools to create many bilingual programs for the children. I would like to create a space for bilingual conversation and group play. (Participant 4)

E.    Universal environment

Research participants raising disabled children in multicultural families suggest that a universal educational environment should be created in which anyone can access the educational environment regardless of disability. In particular, the study participants requested a barrier-free environment, explaining cases in which school facilities were not available due to inaccessibility. In addition, mothers are expecting a school where their children can study to their heart's content in an environment where they can be educated regardless of their ability to learn.
The following are detailed statements of one interviewee.

I feel like Korea is getting better these days, mainly for people with disabilities. I wish that the school facilities were a little more focused on the disabled. My child is going to kindergarten, and they recently remodeled the kindergarten playground. My child is walking with a walker, and she doesn't think of herself and makes her rim high. I had no idea of my child's disability, and I wish the school had thought ahead and made a door or something like a ramp. But it's already finished, so there's nothing I can do. We are having a conversation with the school about how my kid can get in and out independently. Even children without disabilities have to go through middle school and high school hardships, but our children can't do that. (Participant 2)

3.2.2. Mothers' Expectations for Their Children's Education

A.    Strengthening Korean language education

Even now, some schools and multicultural support centers provide Korean language education to children of multicultural families, but the study participants wanted more opportunities to receive Korean language education from their children's schools.
The following are detailed statements of three interviewees.

We want our children to have the basic linguistic competencies to learn other subjects by giving priority to Korean language education over other subjects. To support as a multiculturalist, you have to solve the language first. (Participant 1)

I think there should be a separate teacher who can teach Korean language for special education children. It's not all because you have to have children in your class, but it's necessary to go out for 1–2 h to learn the subject and learn the [Korean] language through it. I think it [personal success] all depends on the language. I think language is the most important thing. And for Korean students with different grades, there is a saying, 'You have to go up to a certain level in your second year', but multicultural students do not speak the language well, so you need to take care of them. (Participant 2)

Studying is the most difficult. School is fun. I hate the study part. So, it is my personal wish. Studying at school is important anyway, but my child also goes to meet social friends, so I think this is also a very important part. However, I feel that the Korean system is too biased towards studying. Like I said, I feel like it's too hard on the study side. (Participant 5)

B.    Special education teachers and their multicultural competencies

In today's multicultural era, the number of multicultural families is increasing, and many children from multicultural families are positioning themselves as consumers of education. As the characteristics of students change, the study participants argue that the educational capabilities of special education teachers who guide and instruct multicultural students should be strengthened. Education authorities should take measures so that not only special education teachers but also integrated special education teachers and school administrators can receive on-the-job training that can improve their multicultural education capabilities on a regular basis.
The following are detailed statements of one interviewee.

There is great lack of education for teachers of children with disabilities. When teachers teach, they have no awareness of what children with disabilities lack. (The special education teacher here is a general special education teacher) So, from my point of view, inclusive means how to teach people with disabilities to live with others rather than teaching such students separately. Normal teachers don't know how to treat children. I have no intention of changing the way I teach (teacher) to include children. (Participant 2)

C.    Differentiated learning

Although it is said that public education is taught according to students' levels, this statement does not meet the expectations of mothers of multicultural children with disabilities. After thoroughly diagnosing and analyzing the strengths and weaknesses of their children by subject, participants are looking forward to an education in which the individualized education program (IEP) is structured to satisfy the students and their parents. Together they are the consumers of education, and the IEP is their plan.

The following are detailed statements of one interviewee.

Even non-disabled children learn differently. There are children who are good [at school] and children who are not good at it, and how I will teach them for children who do not think the same and do not think the same. But Korea doesn't have that level. The idea that everyone should follow the same educational program has to change from the top. This way, the children can adjust their class difficulty and enter a class, even if there is a slight difference, even if there is a difference within the same subject within a class, they can be together. But in Korea, we all have to do the same thing and we have to do the same study together. I think there should be more levels of subject mastery. (Participant 5)

D.    Establishment of a support system

The study participants, having had years of experience as consumers of Korean education, identified that Korean education was highly dependent on private education. Students who lack the ability to study have no support and no chance of success. It is said that there is a need for a support system that can help students learn by organizing instructional aids on a regular basis as is done in other countries.

The following are detailed statements of three interviewees.

To be honest, it seems like a very big problem, and the system itself seems too biased towards private education, so it is not going to be easy. But in my view, for public education to be fair especially for children with no financial resources or children with disabilities, there must be a perception that they do not receive private education at all. So, you have to think that the kids don't know a lot. Even if I say that the system seems to be fixed, the teacher should help a child through one-on-one intervention if he or she is a little weak or falling behind. But there is no such thing. I don't think there's room for that in the system either. (Participant 1)

Korea does not have a diverse culture, so I understand it myself. If multicultural children are involved, I don't think there is a support system. I'm sorry I keep comparing it to Australia, but when other English-speaking children come in, there is a separate class, so I only teach English. I go there for an hour a day to study, or an English as a Second Language teacher comes along and helps my child to be active in class. But in Korea, it is very difficult [to be active] if you do not know the language. I have to go to school, but there is no support and there is no chance of success. (Participant 2)

You have to understand that you don't get good grades because you don't speak the language, and socially, you may not be able to make friends very well. Some classes have a buddy system to assist students who need help. However, it's hard for one classmate to continue doing this all the time, so ask a different person every week. You need to provide help while your child gets used to it. (Participant 5)

E.    Developing an alternative evaluation system

The study participants said that the evaluation method in Korea is uniform. Their view is that the educational goals and evaluation methods are the same without considering the abilities, strengths, and limitations of individual students. Rather than a knowledge-based evaluation method, they wanted to know how to evaluate the emotional and physical

aspects of academic performance. They also wanted to develop an alternative evaluation method that can be applied to children with disabilities as soon as possible.

The following are detailed statements of one interviewee.

> You have to be considerate of your children. You have to look at everything. These children need to see their grades, look at all aspects of their school lives, and think about the difficulties they face. (Participant 2)

F.    Vocational education reinforcement

The study participants who are parents of children with disabilities showed much interest in their children's future. In particular, there are high expectations for vocational education that allows students to find employment after graduating from school rather than for knowledge-oriented education that leads to university. It is hoped that schools will pay more attention to helping people with disabilities to have a high-quality job in an economic reality where their quality of work is often low.

The following are detailed statements of one interviewee.

> They say there are a lot of them when they go to high school. I heard about these opportunities from the teacher. So, if you gave me a little more information, my child could get a job later. But looking at the job positions, I saw that children with disabilities get a part-time, temporary job such as a contract worker. Things like that. So, for these children, someone next to them has to act as a protector, but the country itself should act as a protector, yet that doesn't happen. (Participant 1)

G.    Curricular and extra-curricular activities

It is said that children will be happier to attend school if a teacher can stimulate their interest rather than having them sit in a chair all day studying. The same is true if a teacher conducts classes with a variety of interesting topics rather than focusing on rigid textbooks. Looking at the appearance of high school students in Korea, female marriage immigrant mothers think of the future of their children and feel frustrated.

The following are detailed statements of three interviewees.

> Even if I go to a special class, it doesn't help much because I focus on textbooks. Special children have to do activities such as touching them in various ways, but I don't think the style is suitable for special children because they focus on textbooks. I have taught a lot in Australia, but in Australia, many public schools do not focus on textbooks. They use a variety of activities. But in Korea, children in special education classes will have a harder time because it [instructional methodology] is fixed. Children with special needs especially have a hard time concentrating. I think it's hard to sit down and concentrate for an hour. (Participant 2)

> I would like to encourage children to do what they love to do at school rather than just study. (Participant 3)

> One of my children has a simple physical problem and the other is on the edge of a little learning disability. I myself have a physical mobility issue, but it's not too bad since I can walk alone. My children are socially okay, but they have a hard time studying. It's really hard for them to study so very much. (Participant 5)

## 4. Discussion

This study deeply explored the experiences of marriage immigrants in Korean who are the main caretakers of children with disabilities in multicultural families. First, several viewpoints related to social and emotional hardship were presented. Although the difficulties faced by parents raising children with disabilities are great, raising children with disabilities in multicultural families is undoubtedly more difficult. The results of the study show that social support for multicultural children with disabilities and their families—who are in the blind spot of special education services and social welfare services—is

desperately needed. Practitioners and policy-makers who work with South Korean families and families from similar backgrounds should be understanding and sensitive to such cultural values and challenges in help-seeking behaviours (Oh and Lee 2009). Mothers raising children with disabilities in multicultural families were significantly more likely to receive physical therapy, personal care assistance, nursing services, and legal services (Magana and Hovey 2002).

Second, multiple themes suggested inconvenience related to schools. Although raising children has traditionally been seen as a rewarding task for parents across time, mothers of children with disabilities have particular challenges. The mothers expressed discomfort that their children did not get differentiated Korean language education in a considerate manner. Due to this academic environment, it is said that there is a lack of a social support system for insufficient education and social adaptation. In addition, as time went by, these conditions failed to improve, and consequently educational trust in educational authorities collapsed. Immediately from their children's birth or diagnosis, mothers must contend with the mental and emotional stress of a present disability

Multicultural families also their have their set of unique challenges. Notably there will be two or more cultures present within the family, and how parents choose to navigate the differences will affect their married life and parenting styles. Some studies have shown that certain cultures delegate tasks exclusively to one parent or the other. Female members of married couples often hold the primary responsibility to manage the house and raise the children.

This study's results confirms and complicates extant findings related to mothers of children with disabilities and multicultural families. For instance, many participants noted how they felt emotional distress when attempting to access services for the children with special needs in Korea. Also, female marriage immigrants tend to primarily care for their children, but such responsibilities are rendered more difficult by language barriers—such as a parent-teacher conference–that requires a husband's assistance.

The one place that mothers raising children with disabilities in multicultural families can rely on for their children's education must be none other than the school. This author, born and raised in Korea, has the conviction that expectations for healthy and successful lives can be fulfilled through educational institutions. However, when this study's participants encounter again at school the discomfort they feel in society, the resulting frustration is great. In particular, mothers argued that when children receive instruction as multicultural children with disabilities, the desired education is not being provided because the support system suitable for the child's ability and level is not currently in place. Some participants notably indicated that they decided to return to their home country, saying that they had nothing to expect from Korean education. It has been a long time since Korea entered a multicultural society. The efforts of the state, local governments, and education authorities are expected to create a society where many foreigners and migrants who want to relocate to Korea live happy and productive lives and where their children can go to school safely. Experiences of social discrimination and experiences of career counseling were significantly associated with needs of the Korean language education in children in multi-cultural families (Lee and Byeon 2015). And the policies favoured inclusion and educational planning based on individual needs, and thus implied that students with developmental disabilities would have opportunities for second language learning and (Magana and Hovey 2002).

A third major thematic category involves mother's expectations for Korean society. As marriage immigrants, the study's participants are demanding child care services and bilingual services for the healthy development and rearing of their children. As parents of children with disabilities in multicultural families, they expect more attention and support from the state and local governments in raising their children. In particular, among children and adolescents who grew up in multicultural families, they demanded support for out-of-school children and adolescents who were staying outside of the schooling system because their children did not adjust well to school life and the school could not properly support

them. All parents have expectations of their child's heath and can either agree or disagree about the etiology, diagnosis, treatment and standards of progress (Welterlin and LaRue 2007). Measures of hardship included food insecurity, housing instability, health care access, and telephone disconnection (Parish et al. 2008).

The final thematic category concerned linguistic and educational matters: strengthening Korean language education, special education teachers and their multicultural competencies, differentiated leaning, establishment of support systems, developing an alternative evaluation system, vocational education reinforcement, and curricular and extra-curricular activities. This study's participants demanded strong education and state responsibility for assistance in raising their children. The reason they send their children to school is based on a strong trust in the school. However, their expectations for Korea's public education system, experienced indirectly through their children, have not increased over time. Students' ability levels and learner characteristics are diverse and rapidly changing; furthermore, the reality is that schools and special education teachers who are in charge of instruction are unable to properly respond to the demands and characteristics of their charges requiring special needs. To improve, it is first necessary to listen to the voices of students and parents to deliver education that considers the level and interest of the students. Companies that make mobile phones and electronic devices are releasing a variety of new products within a year; yet, it is not easy to find that the Korean schools educating a democratic populous have innovated to such an extent even after several years. If plans and policies were to change, they still could not overtake the speed of change in the demands of consumers and the social environment. One way to increase the chances that immigrant families and individuals feel comfortable with a professional partnership is for them to seek professionals that are of the same cultural or linguistics background (Welterlin and LaRue 2007). The official service is the implementation of multicultural awareness improvement education for all citizens and the enforcement of the Multicultural Discrimination Act to face the daily hardships. In addition, informal services should be provided with a counseling support system for multicultural families and Korean language education services through volunteers. All citizen must pay a little more attention to these children and local governance must build a good policy for supporting them. In particular, policymakers on social welfare in Korea make the following efforts for multicultural migrant women. A system should be developed to help multicultural migrant women adapt well to Korea. In addition, a specialized institution should be established to consult on children's education. And policymakers on education in Korea as an effort for children, first, multicultural special education experts should be fostered and placed in the school field. Second, it is necessary to create an universal environment in which children with multicultural disabilities can learn without discrimination. This study shows that the essence of multicultural motherhood of children with disabilities leads heavily into advocacy. Future directions in this field should endeavor to create stronger alliances among parents, educational institutions, and government programs so that needs can be met and hope for better futures can be crystalized. But the limitation of this study is that the research results will not be applied to other regions because it targeted fewer cases.

**Funding:** This work was supported by Youngsan University Research Fund of 2021.

**Institutional Review Board Statement:** The study was conducted in accordance with the Declaration of Helsinki.

**Informed Consent Statement:** Informed consent was obtained from all participants involved in the study.

**Data Availability Statement:** Data from this study were not generated available for public use.

**Conflicts of Interest:** The author declares no conflict of interest.

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
