# Peer review of "A Study on Parenting Experiences of Multicultural Families with Disabled Children in Korea"

_socsci, doi:10.3390/socsci11090381_

Round 1

Reviewer 1 Report

Dear authors,

The paper A study on parenting experiences of multicultural families with disabled children in Korea presents a relevant issue. 

However, I would suggest you review the manuscript with additional exploration on the topic in both, the literature and the empirical data aiming density and to support policy with science. Please, see the comments and suggestions below.

In the abstract, the first sentence must be rewritten. As well as other lines e.g., 350 or 561.

A solid theoretical approach is expected, which one did you mobilize? Migration process? Resilience? About literature, as said before, I would suggest presenting clearly the experiences of being a mother of a child with disabilities, and the challenges of being in a multicultural family to later match them with the results of the study. It will lead to providing sufficient background to support the discussion, as you aim “this study deeply explored […]” (line 551).

About the methodology, which literature supported the research design/ decisions? A qualitative case study undertakes the exploration through a variety of lenses to reveal multiple facets of the phenomenon (material gathered in several different ways). Seems to me that the questions in the interviews were very few, it could deeply collect data for a theoretical discussion, including formal and informal support to face the daily hardships. It could relate to the sociodemographic characterization of the participants, it would be interesting to present what education and support they have. And so on.

The analysis could the richer, empirical quotes seems too long for the presentation of the results, e.g., there are 2.5 lines written and inherent quotes: 20 lines. Please, treat that interesting empirical data and get solid results. For example, how this study could support policymakers in providing home-school support for these children/ mothers? This topic is presented in line 299 without exploration, and it is presented later with a couple of subcategories.

A conclusion could be provided to systemize the study’s results.

Success.

All the best. 

Author Response

Reviewer #1:

Comment 1:

In the abstract, the first sentence must be rewritten. As well as other lines e.g., 350 or 561.

Response: Thank you for the constructive comments. I rewrote the sentences indcated. They are provided at the following lines ­in the revised manuscript.

Lines 9-10

This paper employs a qualitative case study to investigate the parenting experiences of multicultural families with disabled children in Korea.

Lines 337-340:

Some participants indicated their desire for national or local governments to create gatherings for mothers in the similar life situations. There they could meet each other to exchange information and to support and encourage each other.

Lines:

(Sentence deleted in revision.)

Comment 2:

I would suggest you review the manuscript with additional exploration on the topic in both, the literature and the empirical data aiming density and to support policy with science.

Response: Thank you for the constructive comments. I added additional literature and empirical data and endeavoured to support policy recommendations with science. This is provided in lines 529-534 ­in the revised manuscript. It now reads:

Practitioners and policy-makers who work with South Korean families and families from similar backgrounds should be understanding and sensitive to such cultural values and challenges in help-seeking behaviors (Oh & Lee, 2009). Mothers raising children with disabilities in multicultural families were significantly more likely to receive physical therapy, personal care assistance, nursing services, and legal services (Magana et al., 2002).

Comment 3:

I would suggest presenting clearly the experiences of being a mother of a child with disabilities, and the challenges of being in a multicultural family to later match them with the results of the study.

Response: Thank you for this comment. I made some changes that would more clearly highlight the experiences of being a mother of a child with disabilities and the challenges of being in a multicultural family. In Lines 607-619 of revised manuscript, it now reads:

The official service is the implementation of multicultural awareness improvement education for all citizens and the enforcement of the Multicultural Discrimination Act to face the daily hardships. In addition, informal services should be provided with a counseling support system for multicultural families and Korean language education services through volunteers. All citizen must pay a little more attention to these children and local governance must build a good policy for supporting them. In particular, policymakers on social welfare in Korea make the following efforts for multicultural migrant women. A system should be developed to help multicultural migrant women adapt well to Korea. In addition, a specialized institution should be established to consult on children's education. And policymakers on education in Korea as an effort for children, first, multicultural special education experts should be fostered and placed in the school field. Second, it is necessary to create an universal environment in which children with multicultural disabilities can learn without discrimination.

Comment 4: Thank you for this comment. In Lines 112-127 of revised manuscript, it now reads:

Seems to me that the questions in the interviews were very few, it could deeply collect data for a theoretical discussion, including formal and informal support to face the daily hardships. It could relate to the sociodemographic characterization of the participants, it would be interesting to present what education and support they have.

Response:

The interview data generated from the study’s five questions created a limited sociodemographic overview of the participants.  Most mothers come from Asian countries geographically close to Korea, and many came to Korea in their 20s. Generally, they had their children within a few years of their arrival.  All mothers have lived in Korea for more than ten years, and currently they are living in a medium-sized city. Their elementary school-aged children’s disabilites are primarily non-physical.  Their specific educational backgrounds are unknown, but the data seems to indicate that they have completed the equivalent of a high school diploma.

Although female marriage immigrants face hardships in obtaining jobs, engaging with society, and communicating in Korean, they do have some measure of support. At the national level, the 2013 Multicultural Families Support Act, 2014 Protection and Promotion of Cultural Diversity Act provide enforceable legal protections against unwarranted hardships. Additionally, the mothers in this study have access to social services through local welfare centers as well as medical services through national health insurance plans. It remains to be seen, though, if their access to such services is quick and easy.

Comment 5:

The analysis could the richer, empirical quotes seems too long for the presentation of the results, e.g., there are 2.5 lines written and inherent quotes: 20 lines. Please, treat that interesting empirical data and get solid results.

Response: Thank you for the creative comments. I am trying to correct according to your comment. The revised section in lines 181–189reads:

  1. Personal guilt about giving birth to a child with a disability

My child was assessed as being in grade 3, and we also found out she has ADHD. We don't want other children to suffer because of our children. We decided to try a drug treatment. There was a noticeable effect: my child takes the medicine and then becomes extremely quiet. When my child is sick, she loses her energy, and all I see her doing is lying down and sitting still. I thought it was okay to stop taking the medicine, so I consulted with the doctor in charge. It was difficult for her to keep taking psychiatric medication for more than 2 years. It hurts my heart to think that my child has a disability because of me. As a parent I wonder what will happen to my child with a disability when I die? It hurts me deeply to think about this. (Participant 1)

The revised section in lines 238–243reads:

  1. Social perspectives on people with disabilities

To be honest, there are a lot of things that the neighborhood welfare centers do. I do think that services for people with disabilities are quite lacking. It would be nice if schools or non-disabled people could broaden their minds about people with disabilities, but I don't think there's a way to solve that. Here (in Korea), people with disabilities are classified as completely disabled; they work separately, so they are on their own. I think they don't find their place in society because they live separately from it. (Participant 2)

The revised section in lines 275–285reads:

  1. Lack of a support system

First, you need to give children support. Espeically language support, as much as possible, with high expectations. In fact, in the long run, your child should get good grades in school. Even if you go to a good college or not, eventually you have to become an adult and get a job in society to make a living. To help with that, looking at the longterm is important. (Participant 1)

Give lots of language support when they are young, Lower their expectations so that they can succeed at their own level. After that there should be a, for example, program for new high school students that allows you to gain other skills when preparing for college. All with lower expectations than the general public. We have to see this plan to the end. If this child fails in school and if there are no future opportunities, then he enters society as a burden to us. (Participant 2)

The revised section in lines 291–298reads:

  1. Distrust of the Korean education system

I wanted the education to be conducted with a focus on the students’ interests rather than on an environment where students only sit on chairs. From our point of view as parents, we lack social skills (the ability to making friends), so at that time it would be better for the child to be in the class. And frankly, helping my child out one-on-one with homework for the purpose of going out and studying (for a special class), it doesn't feel very helpful. I think homeschooling would be better from an educational point of view. Or children can learn through private tutoring (hagwon), but honestly, I don't think schools do anything. (Participant 2)

The revised section in lines 384–392reads:

  1. Universal environment

I feel like Korea is getting better these days, mainly for people with disabilities. I wish that the school facilities were a little more focused on the disabled. My child is going to kindergarten, and they recently remodeled the kindergarten playground. My child is walking with a walker, and she doesn’t think of herself and makes her rim high. I had no idea of ​​my child's disability, and I wish the school had thought ahead and made a door or something like a ramp. But it's already finished, so there's nothing I can do. We are having a conversation with the school about how my kid can get in and out independently. Even children without disabilities have to go through middle school and high school hardships, but our children can't do that. (Participant 2)

The revised section in lines 402–408reads:

  1. Strengthening Korean language education

I think there should be a separate teacher who can teach Korean language for special education children. It's not all because you have to have children in your class, but it's necessary to go out for 1-2 hours to learn the subject and learn the [Korean] language through it. I think it [personal success] all depends on the language. I think language is the most important thing. And for Korean students with different grades, there is a saying, ‘You have to go up to a certain level in your second year’, but multicultural students do not speak the language well, so you need to take care of them. (Participant 2)

The revised section in lines 424–429reads:

  1. Special education teachers and their multicultural competencies

There is great lack of education for teachers of children with disabilities. When teachers teach, they have no awareness of what children with disabilities lack. (The special education teacher here is a general special education teacher) So, from my point of view, inclusive means how to teach people with disabilities to live with others rather than teaching such students separately. Normal teachers don't know how to treat children. I have no intention of changing the way I teach (teacher) to include children. (Participant 2)

The revised section in lines 438–446reads:

  1. Differentiated learning

Even non-disabled children learn differently. There are children who are good [at school] and children who are not good at it, and how I will teach them for children who do not think the same and do not think the same. But Korea doesn't have that level. The idea that everyone should follow the same educational program has to change from the top. This way, the children can adjust their class difficulty and enter a class, even if there is a slight difference, even if there is a difference within the same subject within a class, they can be together. But in Korea, we all have to do the same thing and we have to do the same study together. I think there should be more levels of subject mastery. (Participant 5)

The revised section in lines 507–514reads:

  1. Curricular and extra-curricular activities

Even if I go to a special class, it doesn't help much because I focus on textbooks. Special children have to do activities such as touching them in various ways, but I don't think the style is suitable for special children because they focus on textbooks. I have taught a lot in Australia, but in Australia, many public schools do not focus on textbooks. They use a variety of activities. But in Korea, children in special education classes will have a harder time because it [instructional methodology] is fixed. Children with special needs especially have a hard time concentrating. I think it's hard to sit down and concentrate for an hour. (Participant 2)

Reviewer 2 Report

Thank you for the effort in conducting this interesting study. This study examined the experience of mothers raising children with disabilities in multicultural families with the following research question: “What is the experience of a mother in a multi-cultural family who is raising a child with disabilities in Korea?” I have the following remarks for authors’ consideration: 1) The word “disable” has to be used in a more prudent manner in this paper. This paper, titled “A study on parenting experiences of multicultural families with disabled children in Korea”, has not specified the type of disability being investigated. According to Table 1 in page 3, it seems that the authors were examining the situation of families with children with intellectual disabilities. As a specific form of disability, authors should first discuss the similarities and differences between different forms of intellectual disabilities and physical disabilities. People with disabilities should not be regarded as being in a homogeneous group. That being said, what were the different levels of mental retardation of children in this study? How would that affect participants’ experiences? In some nations, autism is considered as an illness instead of a form of disability. Please elaborate your assumptions. 2) Participants mentioned a lot about existing policies and service support to individuals with disability. These policies and services should be discussed in detail in the literature review section. What are the existing policy and service gaps? Readers who are not residing in Korea might have no idea about that at all. How would the authors compare the existing policies and services in Korea with other developed countries? 3) In the “Results” part, interview excerpts being quoted were too lengthy. Authors should focus more on their own analysis instead of quoting directly from participants in each section. The structure of this part should be significantly reworked. 4) This discussion section is acceptable but a bit sketchy. Please have it linked more closely with discussion about existing policy and service gap in the literature review section (as suggested above). A conclusion section is missing and please also indicate the limitations of this study.

Author Response

Thank you for your excellent commnets. Your great advices made me to imorive my accdemic competence.

Comment 1:

Add page on Korean social agencies response to supporting mothers of children with disabilities.

Response: Thank you for the constructive comments. I am trying to correct according to your comment. The revised section in lines 607–619reads:

 The official service is the implementation of multicultural awareness improvement education for all citizens and the enforcement of the Multicultural Discrimination Act to face the daily hardships. In addition, informal services should be provided with a counseling support system for multicultural families and Korean language education services through volunteers. All citizen must pay a little more attention to these children and local governance must build a good policy for supporting them. In particular, policymakers on social welfare in Korea make the following efforts for multicultural migrant women. A system should be developed to help multicultural migrant women adapt well to Korea. In addition, a specialized institution should be established to consult on children's education. And policymakers on education in Korea as an effort for children, first, multicultural special education experts should be fostered and placed in the school field. Second, it is necessary to create an universal environment in which children with multicultural disabilities can learn without discrimination.

Reviewer #3

Comment 1:

As a specific form of disability, authors should first discuss the similarities and differences between different forms of intellectual disabilities and physical disabilities. People with disabilities should not be regarded as being in a homogeneous group. That being said, what were the different levels of mental retardation of children in this study? How would that affect participants’ experiences? In some nations, autism is considered as an illness instead of a form of disability. Please elaborate your assumptions.

Response: Thank you for the constructive comments. I am trying to correct according to your comment. The revised section in lines 607–619reads:

- In this study, the disabled are limited to those with developmental disabilities. In Korea, the category of developmental disability is regarded as intellectual disability and autistism according to the Developmental Disability Support Act. In this study, the area of people with developmental disabilities is limited to intellectual disabilities and autism disorders. Intellectual disabilities and autism disorders are considered disabilities, not diseases, in Korea. Intellectual disability and autism disorder are types of cognitive disability, and unlike people with physical disabilities, they have cognitive limitations and difficulty in adaptive behavior. The common point between cognitive disabled and physically disabled is that they require environmental changes such as reasonable acccommodation, and that special education services can be received during school age as special education subjects.

Comment 2:

Participants mentioned a lot about existing policies and service support to individuals with disability. These policies and services should be discussed in detail in the literature review section. What are the existing policy and service gaps? Readers who are not residing in Korea might have no idea about that at all. How would the authors compare the existing policies and services in Korea with other developed countries?

Response: Thank you for the constructive comments. I am trying to correct according to your comment. The revised section in lines 607–619reads

Practitioners and policy-makers who work with South Korean families and families from similar backgrounds should be understanding and sensitive to such cultural values and challenges in help-seeking behaviours(Oh & Lee, 2009). Mothers raising children with disabilities in multicultural families were significantly more likely to receive physical therapy, personal care assistance, nursing services, and legal services.(Magana et al., 2002).

                               Second, multiple themes suggested inconvenience related to schools. Although raising children has traditionally been seen as a rewarding task for parents across time, mothers of children with disabilities have particular challenges. The mothers expressed discomfort that their children did not get differentiated Korean language education in a considerate manner. Due to this academic environment, it is said that there is a lack of a social support system for insufficient education and social adaptation. In addition, as time went by, these conditions failed to improve, and consequently educational trust in educational authorities collapsed. Immediately from their children’s birth or diagnosis, mothers must contend with the mental and emotional stress of a present disability

                                         Multicultural families also their have their set of unique challenges. Notably there will be two or more cultures present within the family, and how parents choose to navigate the differences will affect their married life and parenting styles. Some studies have shown that certain cultures delegate tasks exclusively to one parent or the other. Female members of married couples often hold the primary responsibility to manage the house and raise the children.

                              This study’s results confirms and complicates extant findings related to mothers of children with disabilities and multicultural families. For instance, many participants noted how they felt emotional distress when attempting to access services for the children with special needs in Korea.  Also, female marriage immigrants tend to primarily care for their children, but such responsibilities are rendered more difficult by language barriers—such as a parent-teacher conference--that requires a husband’s assistance.

The revised section in lines 607–619reads

And the policies favoured inclusion and educational planning based on individual needs, and thus implied that students with developmental disabilities would have opportunities for second language learning(Magana et al., 2002).

Comment 2:

In the “Results” part, interview excerpts being quoted were too lengthy. Authors should focus more on their own analysis instead of quoting directly from participants in each section. The structure of this part should be significantly reworked.

Response: Thank you for the creative comments. I am trying to correct according to your comment. The revised section in lines 191–199reads:

  1. Personal guilt about giving birth to a child with a disability

My child was assessed as being in grade 3, and we also found out she has ADHD. We don't want other children to suffer because of our children. We decided to try a drug treatment. There was a noticeable effect: my child takes the medicine and then becomes extremely quiet. When my child is sick, she loses her energy, and all I see her doing is lying down and sitting still. I thought it was okay to stop taking the medicine, so I consulted with the doctor in charge. It was difficult for her to keep taking psychiatric medication for more than 2 years. It hurts my heart to think that my child has a disability because of me. As a parent I wonder what will happen to my child with a disability when I die? It hurts me deeply to think about this. (Participant 1)

The revised section in lines 248–253reads:

  1. Social perspectives on people with disabilities

To be honest, there are a lot of things that the neighborhood welfare centers do. I do think that services for people with disabilities are quite lacking. It would be nice if schools or non-disabled people could broaden their minds about people with disabilities, but I don't think there's a way to solve that. Here (in Korea), people with disabilities are classified as completely disabled; they work separately, so they are on their own. I think they don't find their place in society because they live separately from it. (Participant 2)

The revised section in lines 285–295reads:

  1. Lack of a support system

First, you need to give children support. Espeically language support, as much as possible, with high expectations. In fact, in the long run, your child should get good grades in school. Even if you go to a good college or not, eventually you have to become an adult and get a job in society to make a living. To help with that, looking at the longterm is important. (Participant 1)

Give lots of language support when they are young, Lower their expectations so that they can succeed at their own level. After that there should be a, for example, program for new high school students that allows you to gain other skills when preparing for college. All with lower expectations than the general public. We have to see this plan to the end. If this child fails in school and if there are no future opportunities, then he enters society as a burden to us. (Participant 2)

The revised section in lines 301–308reads:

  1. Distrust of the Korean education system

I wanted the education to be conducted with a focus on the students’ interests rather than on an environment where students only sit on chairs. From our point of view as parents, we lack social skills (the ability to making friends), so at that time it would be better for the child to be in the class. And frankly, helping my child out one-on-one with homework for the purpose of going out and studying (for a special class), it doesn't feel very helpful. I think homeschooling would be better from an educational point of view. Or children can learn through private tutoring (hagwon), but honestly, I don't think schools do anything. (Participant 2)

The revised section in lines 394–402reads:

  1. Universal environment

I feel like Korea is getting better these days, mainly for people with disabilities. I wish that the school facilities were a little more focused on the disabled. My child is going to kindergarten, and they recently remodeled the kindergarten playground. My child is walking with a walker, and she doesn’t think of herself and makes her rim high. I had no idea of ​​my child's disability, and I wish the school had thought ahead and made a door or something like a ramp. But it's already finished, so there's nothing I can do. We are having a conversation with the school about how my kid can get in and out independently. Even children without disabilities have to go through middle school and high school hardships, but our children can't do that. (Participant 2)

The revised section in lines 412–418reads:

  1. Strengthening Korean language education

I think there should be a separate teacher who can teach Korean language for special education children. It's not all because you have to have children in your class, but it's necessary to go out for 1-2 hours to learn the subject and learn the [Korean] language through it. I think it [personal success] all depends on the language. I think language is the most important thing. And for Korean students with different grades, there is a saying, ‘You have to go up to a certain level in your second year’, but multicultural students do not speak the language well, so you need to take care of them. (Participant 2)

The revised section in lines 434–439reads:

  1. Special education teachers and their multicultural competencies

There is great lack of education for teachers of children with disabilities. When teachers teach, they have no awareness of what children with disabilities lack. (The special education teacher here is a general special education teacher) So, from my point of view, inclusive means how to teach people with disabilities to live with others rather than teaching such students separately. Normal teachers don't know how to treat children. I have no intention of changing the way I teach (teacher) to include children. (Participant 2)

The revised section in lines 448–456reads:

  1. Differentiated learning

Even non-disabled children learn differently. There are children who are good [at school] and children who are not good at it, and how I will teach them for children who do not think the same and do not think the same. But Korea doesn't have that level. The idea that everyone should follow the same educational program has to change from the top. This way, the children can adjust their class difficulty and enter a class, even if there is a slight difference, even if there is a difference within the same subject within a class, they can be together. But in Korea, we all have to do the same thing and we have to do the same study together. I think there should be more levels of subject mastery. (Participant 5)

The revised section in lines 517–524reads:

  1. Curricular and extra-curricular activities

Even if I go to a special class, it doesn't help much because I focus on textbooks. Special children have to do activities such as touching them in various ways, but I don't think the style is suitable for special children because they focus on textbooks. I have taught a lot in Australia, but in Australia, many public schools do not focus on textbooks. They use a variety of activities. But in Korea, children in special education classes will have a harder time because it [instructional methodology] is fixed. Children with special needs especially have a hard time concentrating. I think it's hard to sit down and concentrate for an hour. (Participant 2)

Comment 4:

This discussion section is acceptable but a bit sketchy. Please have it linked more closely with discussion about existing policy and service gap in the literature review section (as suggested above). A conclusion section is missing and please also indicate the limitations of this study.

Response: Thank you for the creative comments. I am trying to correct according to your comment. The revised section in lines 617–635reads:

The official service is the implementation of multicultural awareness improvement education for all citizens and the enforcement of the Multicultural Discrimination Act to face the daily hardships. In addition, informal services should be provided with a counseling support system for multicultural families and Korean language education services through volunteers. All citizen must pay a little more attention to these children and local governance must build a good policy for supporting them. In particular, policymakers on social welfare in Korea make the following efforts for multicultural migrant women. A system should be developed to help multicultural migrant women adapt well to Korea. In addition, a specialized institution should be established to consult on children's education. And policymakers on education in Korea as an effort for children, first, multicultural special education experts should be fostered and placed in the school field. Second, it is necessary to create an universal environment in which children with multicultural disabilities can learn without discrimination. This study shows that the essence of multicultural motherhood of children with disabilities leads heavily into advocacy. Future directions in this field should endeavor to create stronger alliances among parents, educational institutions, and government programs so that needs can be met and hope for better futures can be crystalized. But the limitation of this study is that the research results will not be applied to other regions because it targeted fewer cases.

Reviewer 3 Report

Your work is outstanding. Your choice of method is ideal for this type of research.  There are readers who will say the interpretation is biased as a result of personal interview. Those readers won't understand that surveys are more biased: researchers write the questions and  provide responses and ask respondents to "pick an answer."  For those who don't understand your method tell them why you chose interviews instead of a survey and how you analyzed the data (give examples of the process of analysis).  Perhaps add page on Korean social agencies response to supporting mothers of children with disabilities. As a personal note, many decades ago I was awarded a Guggenheim fellowship to study Korean families who "propped off " their children at agencies and churches because they were too poor to feed them.  If you focus on methods, you can revise the manuscript for an anthropology journal.     

Author Response

Comment 1:

Add page on Korean social agencies response to supporting mothers of children with disabilities.

Response: Thank you for the constructive comments. I am trying to correct according to your comment. The revised section in lines 607–619reads:

 The official service is the implementation of multicultural awareness improvement education for all citizens and the enforcement of the Multicultural Discrimination Act to face the daily hardships. In addition, informal services should be provided with a counseling support system for multicultural families and Korean language education services through volunteers. All citizen must pay a little more attention to these children and local governance must build a good policy for supporting them. In particular, policymakers on social welfare in Korea make the following efforts for multicultural migrant women. A system should be developed to help multicultural migrant women adapt well to Korea. In addition, a specialized institution should be established to consult on children's education. And policymakers on education in Korea as an effort for children, first, multicultural special education experts should be fostered and placed in the school field. Second, it is necessary to create an universal environment in which children with multicultural disabilities can learn without discrimination.

Round 2

Reviewer 1 Report

I appreciated the progress.

Reviewer 2 Report

I do not have further suggestions, thank you.